# A surface-stabilized ozonide triggers bromide oxidation at the aqueous solution-vapour interface

Luca Artiglia[1,2], Jacinta Edebeli[1,3], Fabrizio Orlando[1], Shuzhen Chen[1,3], Ming-Tao Lee[1,4], Pablo Corral Arroyo[1,5], Anina Gilgen[1,3], Thorsten Bartels-Rausch[1], Armin Kleibert [6], Mario Vazdar [7], Marcelo Andres Carignano[8], Joseph S. Francisco[9], Paul B. Shepson[10], Ivan Gladich [8] & Markus Ammann [1]

Oxidation of bromide in aqueous environments initiates the formation of molecular halogen compounds, which is important for the global tropospheric ozone budget. In the aqueous bulk, oxidation of bromide by ozone involves a [Br•OOO⁻] complex as intermediate. Here we report liquid jet X-ray photoelectron spectroscopy measurements that provide direct experimental evidence for the ozonide and establish its propensity for the solution-vapour interface. Theoretical calculations support these findings, showing that water stabilizes the ozonide and lowers the energy of the transition state at neutral pH. Kinetic experiments confirm the dominance of the heterogeneous oxidation route established by this precursor at low, atmospherically relevant ozone concentrations. Taken together, our results provide a strong case of different reaction kinetics and mechanisms of reactions occurring at the aqueous phase-vapour interface compared with the bulk aqueous phase.

---

[1] Laboratory of Environmental Chemistry, Paul Scherrer Institut, 5232 Villigen, Switzerland. [2] Laboratory for Catalysis and Sustainable Chemistry, Paul Scherrer Institut, 5232 Villigen, Switzerland. [3] Institute of Atmospheric and Climate Sciences, ETH Zürich, 8092 Zürich, Switzerland. [4] Chemical Physics Division, Department of Physics, Stockholm University, 10691 Stockholm, Sweden. [5] Department of Chemistry and Biochemistry, University of Bern, 3012 Bern, Switzerland. [6] Swiss Light Source, Paul Scherrer Institut, 5232 Villigen, Switzerland. [7] Division of Organic Chemistry and Biochemistry, Rudjer Bošković Institute, Bijenička 54, 10000 Zagreb, Croatia. [8] Qatar Environment and Energy Research Institute, Hamad Bin Khalifa University, P.O. Box 34110, Doha, Qatar. [9] Department of Chemistry, University of Nebraska-Lincoln, 433 Hamilton Hall, Lincoln, NE 68588-0304, USA. [10] Department of Chemistry, and Department of Earth, Atmospheric and Planetary Sciences, Purdue University, West Lafayette, IN 46097, USA. Correspondence and requests for materials should be addressed to I.G. (email: igladich@hbku.edu.qa) or to M.A. (email: markus.ammann@psi.ch)

In atmospheric chemistry, halogen atoms resulting from photolysis of both organic and inorganic halogen compounds[1, 2] are important catalysts for ozone depletion both in the stratosphere and the troposphere, with varying relative roles of chlorine, bromine and iodine in these compartments[3]. Halogen atoms are also oxidants themselves towards organic compounds and are implicated in the deposition of mercury[4, 5]. Furthermore, halogen atoms are intermediates in waste water treatment, where halogenated organic secondary products are of concern[6].

For tropospheric chemistry, the main inorganic route is initiated by the oxidation of aqueous phase bromide to either bromine atom or hypobromous acid (HOBr), which combine with other halide ions to form molecular halogen compounds that are released into the gas phase. Bromide is abundant and sometimes enriched[7] in sea water and thus at the ocean surface, in sea spray particles, in brines associated with sea ice, frost flowers or snow, in artificial or natural salt pans, and in volcanic emissions.

Many radical oxidants such as •OH or excited triplets of organic chromophores require UV or at least near UV light to drive bromide oxidation. Therefore, both in the (dark) polar marine boundary layer and the upper troposphere, ozone ($O_3$) is one of the most important oxidants[5, 8]. The product HOBr ($pK_a$ = 8.65) reacts further in an acid-catalyzed mechanism with chloride, bromide or iodide to form bromine ($Br_2$), BrCl or BrI. The initial formation of HOBr limits the release of halogens to the gas phase, while a complex suite of gas phase and multiphase processes controls the halogen chemistry and the $O_3$ budget later on.

The bulk aqueous phase, acid-catalyzed oxidation of bromide by $O_3$ to HOBr has been studied for a long time due to its relevance in the debromination of waste water[6, 9–14].

$$Br^- + O_3 \rightarrow [Br\bullet OOO^-] \tag{1}$$

$$[Br\bullet OOO^-] + H^+ \rightarrow HOBr + O_2 \tag{2}$$

$$[Br\bullet OOO^-] + H_2O \rightarrow HOBr + O_2 + OH^- \tag{3}$$

$$Net: \; Br^- + O_3 + H_2O \rightarrow HOBr + O_2 + OH^- \tag{4}$$

To explain discrepancies in the apparent kinetics among different experimental observations, Liu et al.[12] proposed there must be an ozone adduct (a steady-state intermediate formed in Eq. 1) with the nucleophile, bromide, prior to oxygen atom transfer with release of molecular oxygen Eqs. 2, 3. The structure of the [Br•OOO⁻] adduct (which we refer to as an "ozonide"), involves a weak bond between the bromide and the oxygen of ozone[12]. The aqueous solvation sphere has a large effect on the stability and reactivity of [Br•OOO⁻]. The kinetic data indicated formation of [Br•OOO⁻] as a steady-state intermediate with an acid-assisted step to form HOBr and molecular oxygen[12]. Further calculations, performed for the gas phase, confirmed the stability of the [Br•OOO⁻] intermediate, showing intricate details of its reactivity during the oxygen atom transfer process[13].

So far, it is often assumed that [Br•OOO⁻] is stabilized by solvation, and that the reaction occurs in the bulk. Furthermore, the rate coefficient for the overall reaction[12] and the low solubility of ozone (0.025 M atm⁻¹ at 273 K)[15] suggests that the formation of HOBr through the bulk aqueous phase route is rather inefficient and would not seem important as a source of gas phase bromine in the environment. In turn, heterogeneous oxidation experiments have consistently shown that oxidation at the aqueous solution-air interface may dominate in environments with high aqueous surface to volume ratios[1, 16–22]. Oldridge and Abbatt[19] used the inverse ozone concentration dependence of the uptake coefficient, $\gamma$, to suggest a Langmuir-Hinshelwood type

process occurring on the surface in parallel to the bulk aqueous phase oxidation (see Supplementary Note 1 for a detailed description of the concept of heterogeneous kinetics of this system)[19], where $\gamma$ is the oxidation rate normalized by the gas collision rate of ozone with the surface.

Linked to a single composition and single temperature, the nature of the surface reaction, the identity of the potential precursor and its preference for the surface have never been tracked down. Similar behaviour of other heterogeneous oxidation processes suggest a general type of ozone intermediate formed on electron rich surfaces of widely differing chemical composition and phase[23–26].

Finally, the idea of efficient surface oxidation of bromide had originally been related to the preference of the larger, more polarizable halide ions for the aqueous solution-vapour interface, as observed in molecular dynamics (MD) simulations[27, 28] and X-ray photoelectron spectroscopy (XPS) experiments on static deliquesced crystals[29]. Recent MD simulations predicting the photoemission signal intensity by means of photoelectron scattering calculations and previous liquid jet XPS data indicate a less pronounced surface enhancement, which is in better agreement with the overall positive surface tension change (and thus negative surface excess) of bromide solutions[30, 31]. Therefore, the surface precursor limited bromide oxidation is likely unrelated to the amount of bromide ions directly at the interface of a neat salt solution.

In this work we use liquid jet XPS, a powerful tool to directly assess the structure of the interfacial region with high chemical selectivity and high selectivity for the interface due to the probe (or information) depth of just a few nanometres[32]. The continuously renewed surface of the flowing liquid jet avoids radiation damage effects but still allows probing a surface that is locally in equilibrium with the first few tens of nanometres of bulk aqueous phase[33, 34] (as explained further in Supplementary Note 4). We combine classical heterogeneous kinetics experiments, which reconfirm the characteristics of the surface reaction, with theoretical investigations (including both the ab initio electronic structure and MD method) and with liquid jet XPS. The results provide strong evidence for the [Br•OOO⁻] intermediate, its preference for the liquid-vapour interface, the effect of water on its stabilization, and the reaction path to products.

## Results

**Kinetic results.** Figure 1 shows the measured ozone uptake coefficients ($\gamma_{obs}$) at 274 K and pH 1 as a function of the square root of the bromide activity (plot a) and as a function of the ozone concentration (plot b). This data set confirms that $\gamma$ decreases with increasing ozone concentration, which cannot be explained by aqueous phase bulk kinetics (more data available in Supplementary Fig. 1). Reactivity in the bulk depends on the solubility of ozone, ionic strength of the solution, and diffusivity of ozone (Supplementary Note 1). Figure 1 is in line with the involvement of a precursor on the surface, whose surface coverage saturates at higher ozone concentrations in the gas phase. These data confirm the findings of Oldridge and Abbatt[19]. At high ozone concentrations, the reactivity is dominated by the bulk aqueous phase reaction, which scales with the square root of bromide activity (Supplementary Note 1, Eq. 4). At lower gas phase ozone concentrations, the surface component becomes relatively higher. Figure 1 also contains fits to the measured uptake coefficients that take into account both components (as explained in Supplementary Note 1 and further below). Notably, the data indicate that this surface component is relatively higher also at lower bromide concentration, which will be discussed further below.

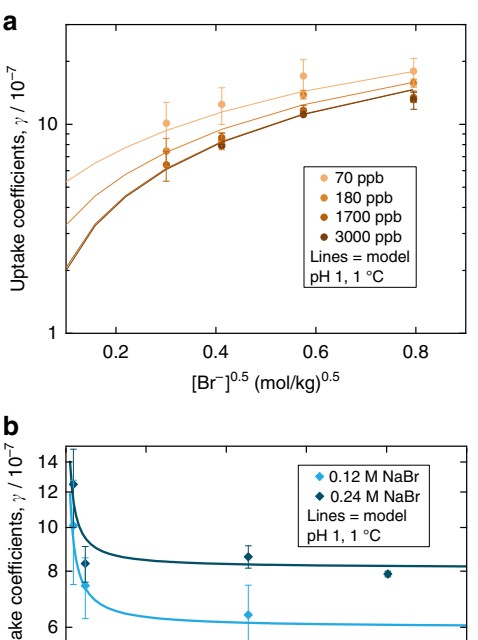

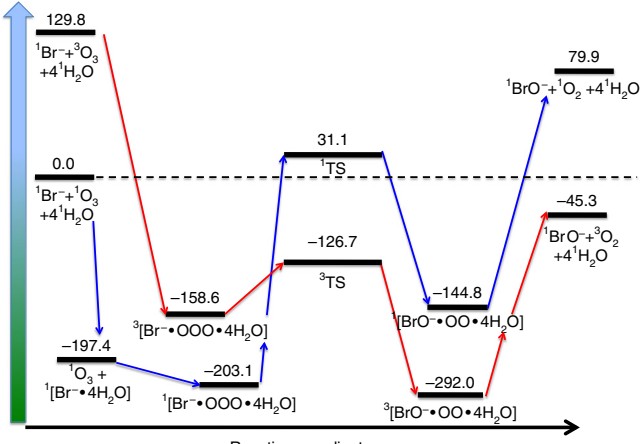

**Fig. 2** Energetic profile for the bromide-ozone reaction with water molecules. The figure reports the energetic profile for the reaction of bromide with ozone in a cluster consisting of four water along the singlet (*blue*) and triplet (*red*) surface. Electronic structure calculations were performed at CCSD(T)/6–311++G(df,p)//MP2/6–311++G(df,p) level. All energies, which include the Zero Point Energy (ZPE) correction, are relative to the singlet reactants and are reported in kJ mol$^{-1}$. The spin crossing between the two potential energy surfaces is highlighted by the intersection of the *red* and *blue arrows* between the pre-reaction complexes and the transition states

**Fig. 1** Heterogeneous kinetics results. Measured uptake coefficients of ozone, $\gamma_{obs}$, at different ozone concentrations (*symbols*) on aqueous bromide solutions, along with calculated uptake coefficients (*lines*) from a parameterized fit to several measured datasets, explained in Supplementary Notes 1 and 2. The uptake coefficient is the loss rate of gas phase ozone to the aqueous solution normalized to the gas kinetic collision rate with the aqueous solution surface. In plot **a**, data from experiments performed with solutions of varying bromide content are shown for four different ozone concentrations in the gas phase, while in plot **b**, data result from experiments, in which the ozone concentration was varied, for two different bromide concentrations in the aqueous phase. *Error bars* represent s.d. of measured values

**Ab initio calculations of [Br•OOO$^-$] stability in water**. Previous electronic structure calculations for the oxidation of bromide by ozone in the gas phase showed that the reaction is activated by the formation of a stable $^1$[Br•OOO$^-$] pre-complex likely followed by a spin crossing to the triplet potential energy surface due to the different spin state of the products[13]. Herein, additional electronic structure calculations were performed to address the influence of water on the stability of the $^1$[Br•OOO$^-$] pre-complex. Water can profoundly affect the reaction rates and the nature of different atmospheric (and non-) reactions by coordinating (hydrogen) bonds to the reagents, products, and the different reaction pre-complexes[35]. Reaction profiles in solvated environment are often remarkably different from the ones obtained in the gas phase[36, 37]. The water cluster approach has been successfully used to model reactions in aqueous solutions and at the surface of water, ice and aerosols[36–42]. If a sufficient number of water molecules are included in the electronic structure calculations, the water cluster approach has been proven to describe the solvation environment of different reaction systems reasonably well[43–45]. In our case, Supplementary Fig. 5 shows how the energy difference between reaction complexes converge within the chemical accuracy of the method[46] for a cluster of four water molecules. The water cluster approach has the advantage of keeping the system size small, allowing the use of high-level theory electronic structure calculations. Figure 2 shows the reaction profile with four water molecules, using the singlet

ground state of the reactants as an initial and reference level for the energetics. Comparing this reaction profile with previous gas phase results, we conclude that water further stabilizes the pre-complex with respect to the reactants and the other reaction complexes.

This stabilization effect is also visible in the optimized geometries of $^1$[Br•OOO$^-$] with water (Supplementary Note 3). The larger the number of water molecules, the closer the distance between bromine and the nearest oxygen atom. Moreover, the energy differences between $^1$[Br•OOO$^-$] and the transition states on the singlet and triplet surfaces are remarkably high. Conventional transition state theory provides a qualitative estimation of the rate constant along the singlet surface of the order of $10^{-28}$ s$^{-1}$. In conclusion, water stabilizes the $^1$[Br•OOO$^-$] pre-complex, while the heights of the transition barriers and the spin-crossing nature of the reaction suggest a slow kinetics, favouring a longer lifetime of the $^1$[Br•OOO$^-$] complex at the liquid water surface than in the gas phase. Next we address the extent to which we can detect this species experimentally.

**Liquid jet XPS experiments**. Figure 3 shows the photoemission signal of the Br 3$d$ core level region, a double-peak structure due to the spin orbit splitting. After normalization to the maximum, the spectra acquired before and during in-situ dosing of oxygen at a pressure of 0.25 mbar are identical. The signal collected while dosing a mixture of oxygen and ozone (~1.0 % ozone, same pressure) displays clear changes both in the 3$d_{5/2}$ and 3$d_{3/2}$ region. This suggests the presence of a second doublet, which is highlighted in purple in Fig. 3c, positively shifted by 0.7 eV with respect to the main peaks assigned to bromide (72.10 and 73.15 eV).

To identify the new spectral feature, we acquired the Br 3$d$ spectra of two reference aqueous solutions of possible oxidation products, i.e., 0.08 M hypobromite and 0.125 M bromate (Fig. 3d). As expected, the higher the oxidation state of bromine, the larger

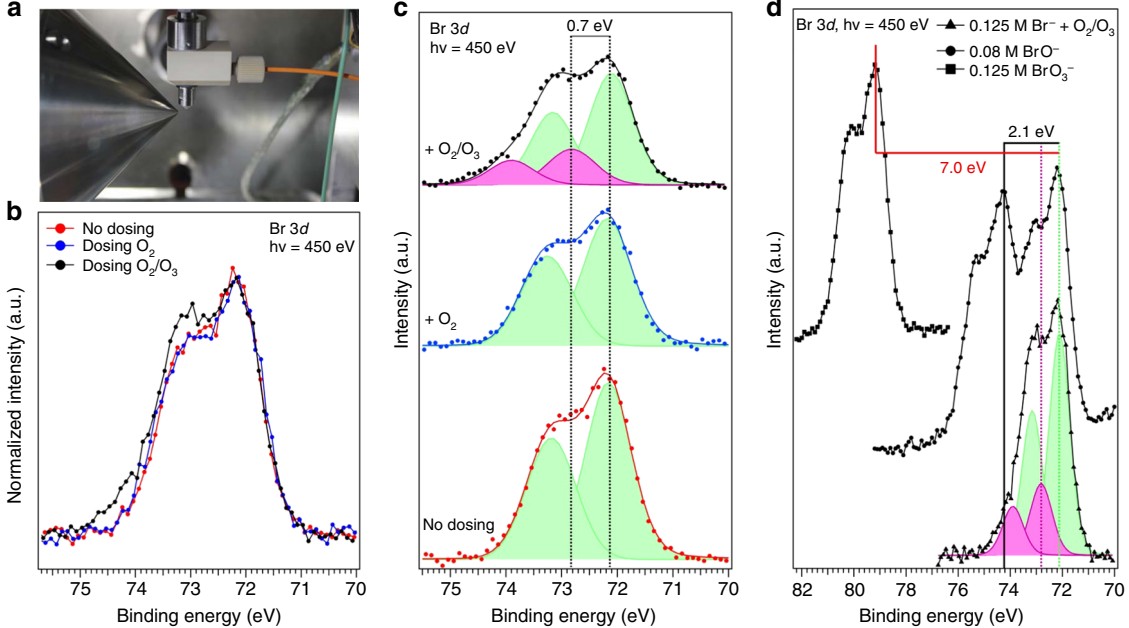

**Fig. 3** Photoemission spectra of the Br 3*d* peak acquired in situ. **a** Picture of the liquid microjet assembly, equipped with the gas dosing system, during operation; **b** Superimposition of the Br 3*d* photoemission spectra, normalized to the maximum, acquired before dosing (*red*), while dosing oxygen (*blue*), and while dosing a mixture of 1% ozone in oxygen; **c** deconvolution of the raw spectra in plot **b**, performed using Gaussian peaks after subtraction of a Shirley background; **d** comparison of the Br 3*d* photoemission spectra acquired while dosing a mixture of 1% ozone in oxygen with two reference spectra of hypobromite and bromate

**Table 1 Theoretical evaluation of the core electron binding energies (CEBE) of the bromide ions, the [Br•OOO⁻] complex, the hypobromite ion in the gas phase, and all ions with three and five water molecules**

|  | CEBE(Br⁻$_{3d}$), eV | Δ(CEBE − CEBE$_{Br⁻}$), eV |
|---|---|---|
| ¹Br⁻ | 70.01$^{(a)}$ | 0.00 |
| ¹[Br•OOO⁻] | 72.79$^{(a)}$ | 2.78 |
| ¹BrO⁻ | 73.10$^{(a)}$ | 3.09 |
| ¹Br⁻ + 3 H₂O | 71.73$^{(a)}$ | 0.00 |
| ¹[Br•OOO⁻] + 3 H₂O | 74.33$^{(a)}$ | 2.60 |
| ¹BrO⁻ + 3 H₂O | 75.33$^{(a)}$ | 3.60 |
| ¹Br⁻ + 5 H₂O$^{(b)}$ | 72.02$^{(b)}$ | 0.00 |
| ¹[Br•OOO⁻] + 5 H₂O$^{(b)}$ | 73.83$^{(b)}$ | 1.81 |
| ¹BrO⁻ + 5 H₂O$^{(b)}$ | 74.69$^{(b)}$ | 2.67 |

Δ(CEBE − CEBE$_{Br⁻}$) is the energy difference (in eV) between the species and the bromide, taken as a reference. CEBE marked with (a) are calculated on the top of MP2 optimized geometries. CEBE marked with (b) are averaged over the values obtained from five different snapshots extracted from the first-principles MD trajectory

the positive shift of the binding energy. A chemical shift of + 2.1 eV is observed for hypobromite, and of + 7.0 eV for bromate. None of them corresponds to that of the new doublet. It is well known that X-rays can induce the radiolysis of water[47], leading to the production of reactive hydroxyl radicals that may react with bromide ions. All the Br 3*d* spectra were recorded under the same experimental conditions (excitation energy, photon flux), and the high speed of the liquid filament ensures that the concentration of photo-generated hydroxyl radicals remains below $1.0 \times 10^{-6}$ mol l⁻¹. Therefore beam damage can be excluded. In parallel to the XPS data, we calculated the core electron binding energy (CEBE) at MP2/aug--cc---pvtz theory level of both the structure obtained at MP2/6---311++(df,p) geometric optimization and from first-principle MD (Table 1). It is important to highlight that the CEBEs are calculated for the species solvated in small

water clusters. This reproduces quite well an interfacial environment but may not reproduce well species that are fully solvated in the bulk. Compared with the CEBEs of gas phase species the Δ between the bromide and the [Br•OOO⁻] decreases when water is added, whereas that between the [Br•OOO⁻] and the hypobromite increases. This suggests that the solvation sphere has a fundamental role. At the same time, theoretical calculations reproduce the same sequence of CEBE observed by XPS, i.e., CEBE (Br⁻) < CEBE([Br•OOO⁻]) < CEBE(BrO⁻). In summary, the combination of in situ XPS and theoretical calculations provides strong indications for the formation of an ozonide complex.

**Surface propensity of the intermediate**. First-principle MD simulations, which are a particularly suitable tool to study the dynamics and stability of non-standard compounds, were used to address the bulk vs. surface propensity of the different reaction intermediates[48]. Figure 4a shows a snapshot (corresponding to 8.5 ps) of the MD trajectory of the pre-complex on the surface of a water slab at 300 K. The inset shows the distance between bromine and each of the oxygen atoms of ¹[Br•OOO⁻] along the MD trajectory. This distance fluctuates around the average value of 2.7 Å, which is consistent with that obtained for the optimized geometries by electronic structure calculations (see ref. 9 and Supplementary Note 3). This further supports the scenario of a pre-complex stabilized on the surface of liquid water. Moreover, Fig. 4b shows the density profile of the Br and OOO groups in the ¹[Br•OOO⁻] intermediate position along the coordinate perpendicular to the water interface, confirming that ¹[Br•OOO⁻] remains at the interface during the whole trajectory, with the Br group close to the OOO group.

In parallel to the MD simulations, we acquired the Br 3*d* photoemission peak at increasing excitation energies, hence, at increasing photoelectron kinetic energy or information depth, as shown in Fig. 4c. A decrease of the relative intensity of the peaks associated to the [Br•OOO⁻] complex is observed at the highest

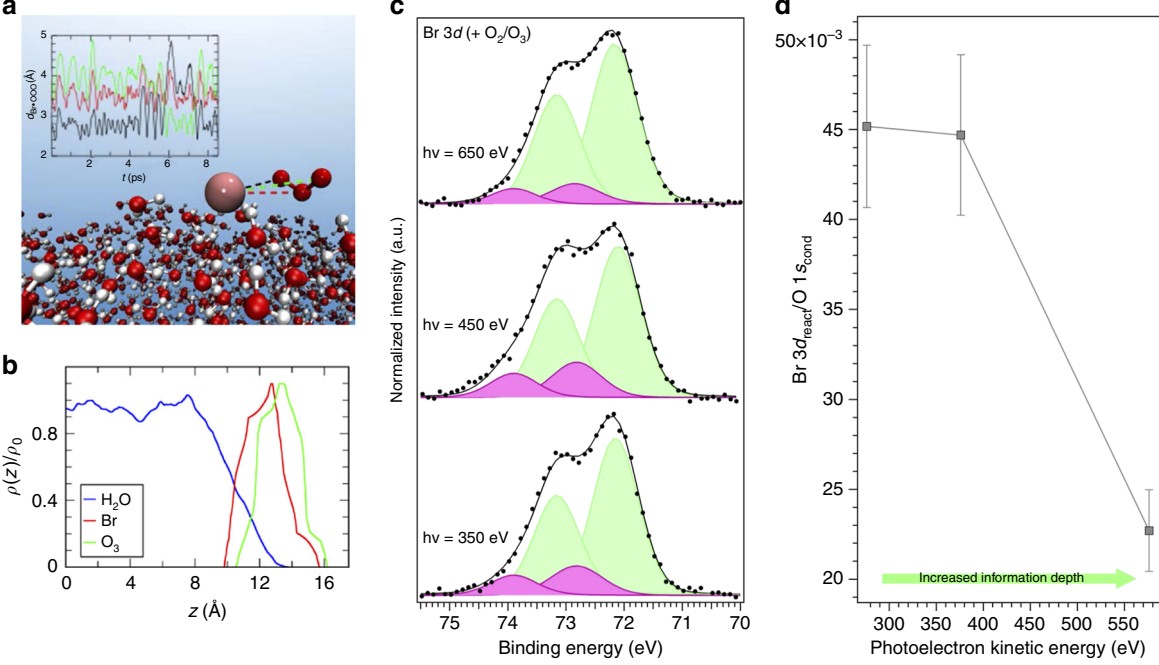

**Fig. 4** Surface propensity of [Br•OOO⁻]. **a** Snapshot from the first-principle MD trajectory demonstrating the stability of [Br•OOO⁻] on the surface of liquid water. The *inset* shows the distances between the bromine and each of the oxygen atoms in the ozone molecule recorded during the 8.5 ps MD trajectory. **b** From the same trajectory, the bromide and ozone density profile showing the position of the centre of mass of these two groups along the coordinate perpendicular to the interface. In *blue*, the water profile as reference in arbitrary units. **c** deconvolution of the Br $3d$ photoemission spectra (performed using Gaussian peaks after subtraction of a Shirley background), normalized to the area, acquired at h$\nu$ = 350, 450, and 650 eV, and corresponding to photoelectron kinetic energies of 276, 376 and 575 eV (Br $3d$ region centroid). **d** Plot of the intensity of the Br $3d$ peaks associated to the [Br•OOO⁻] complex normalized to the O $1s$ (peak acquired with second order light) from the condensed phase (Supplementary Note 4), as a function of the photoelectron kinetic energy. The *error bars* were calculated by propagating the errors associated with each peak area of three independent measurements

excitation energy. We acquired the O $1s$ signal at the same photoelectron kinetic energies (Supplementary Fig. 6a) and used the area of the peak corresponding to the condensed phase to normalize the area of the [Br•OOO⁻] doublet. Figure 4d shows that this ratio, normalized to the cross sections of the elements, decreases considerably at a kinetic energy of 576 eV, i.e., with greater depth sampled. This indicates that the [Br•OOO⁻] complex has a propensity for the surface, in good agreement with the theoretical calculations discussed above. The photo-emission intensity ratio between the complex and the bromide in the bulk ($I_{BrOOO-}/I_{Br-}$) shows a similar behaviour as a function of the photoelectron kinetic energy (Supplementary Fig. 7). An estimate of the surface coverage of the [Br•OOO⁻] complex at the ozone concentration used in this study ($2.5 \times 10^{-8}$ mol l⁻¹), obtained employing different models, yields ~$2.0 \times 10^{12}$ complexes per cm² (Supplementary Note 4 and Supplementary Fig. 7 for more details). In conclusion, both MD results and XPS data confirm that [Br•OOO⁻] resides at the interface, where it is stabilized by interfacial water molecules.

## Discussion

As previously mentioned, the reaction of ozone with bromide in the bulk aqueous phase is fairly slow, with a rate coefficient of around 163 M⁻¹ s⁻¹ at 293 K and neutral pH[11], and 38 M⁻¹ s⁻¹ at 273 K and neutral pH[12]. Therefore, when ozone is dosed, we would not expect the formation of HOBr or hypobromite within the ~ 100 µs exposure between the gas dosing system and the detection point of XPS. In turn, the actual collision rate of ozone molecules with the surface at about $6.0 \times 10^{17}$ molecules per cm² s⁻¹ is sufficient to build up a high surface coverage of the complex with bromide, if the association kinetics are fast

(Supplementary Note 4). This of course also requires that the availability of bromide at the surface is sufficient. In the absence of any pre-existing enhanced concentration on the surface, for-mation of an even maximum conceivable complex surface cov-erage of around $10^{12}$ molecules per cm² (as estimated from the XPS data) would deplete roughly the topmost few nanometres of a 0.1 M solution. Replenishing this by diffusion from the deeper bulk requires a few microseconds (Supplementary Note 4)[21, 32], fast enough to ensure equilibration between the surface and the bulk under the present experimental conditions. Due to the substantial transition state barrier towards products, the steady-state surface coverage establishes quickly, and is determined by the rapid association and dissociation of ozone and bromide.

Combining the theoretical and spectroscopic results, we have extended the bulk phase kinetic mechanism by adding a simple scheme for the surface reaction. The [Br•OOO⁻] replaces the adsorbed precursor of the Langmuir-Hinshelwood mechanism (Supplementary Note 1), which remains in equilibrium with the gas phase. The temperature dependence of the equilibrium constant is driven by the energy difference, as in Fig. 2. Furthermore, we assumed that the energy difference to the transition state on the triplet energy surface (and thus including spin-crossing, which is more likely to occur with heavy elements as bromine) would determine the decomposition kinetics on the surface and the temperature dependence for the formation of products.

As Fig. 1 and Supplementary Fig. 2 show, this parameterization allows reconciling the data presented in this study and those of Oldridge and Abbatt[19] within less than a factor of two, both with respect to the absolute value of the uptake coefficient and the concentration range where the surface reaction dominates. The

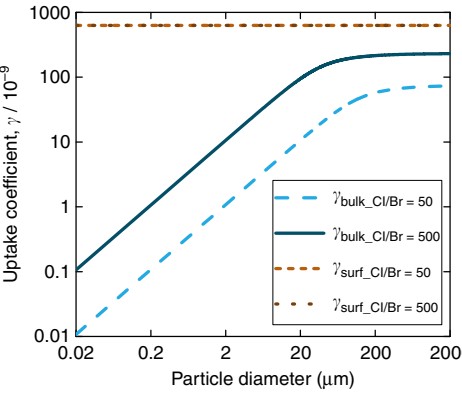

**Fig. 5** Implications for ozone uptake under atmospheric conditions. Predicted uptake coefficient for bulk reaction only (*blue lines*) and uptake coefficient due to surface reaction only (*brown lines*) for deliquesced sea salt solution equilibrated at 80% relative humidity (4.0 M Cl⁻) and a chloride to bromide ratio of 50 (*light blue* and *light brown lines*) and 500 (*dark blue* and *dark brown*)), as a function of the diameter of a spherical brine droplet or aerosol particle. The *brown lines* overlap, because the surface coverage of the bromide ozonide complex, and thus also the surface reaction rate, does not depend on the bromide concentration in the aqueous phase within the relevant range. This plot emphasizes the predominance of a surface process, especially for smaller particle sizes

[Br•OOO⁻] complex is in fast equilibrium with gas phase ozone. Its surface coverage is saturated above $10^{11}$ molecule per cm³, thus both at the lowest ozone concentration of the present study and at all atmospheric levels, and depends very weakly on the temperature. As apparent from Fig. 1, the extent of the surface reaction, manifesting itself in the positive deviation from the bulk phase reactivity with decreasing gas phase concentration, gets relatively larger at lower bromide concentration. This is nicely reproduced by the surface reaction model constructed here. The measured kinetic data do not allow constraining the magnitude of the saturating surface coverage (to compare with that derived from XPS), since no data are available at low enough ozone concentrations to escape the saturating regime. For the present data, the rate of the surface reaction is only constrained by the product of the coverage and the surface reaction rate coefficient. The first order rate coefficient for the decomposition of the complex into hypobromite on the surface as derived from the kinetic data is about $10^{-3}$ s⁻¹.

Under atmospheric conditions, the relative importance of bulk and surface reactivity needs a more careful analysis. In the environment, salt brine may occur in various forms and spanning a large size range in snow, on sea ice, on frost flowers, in salt pans, or as sea spray aerosol particles. Figure 5 shows the bulk reactivity (in absence of surface reaction) and the surface reactivity according to the mechanism in this study, for a hypothetic spherical sea salt brine droplet or aerosol particle as a function of the diameter, for both a bromide concentration as in sea water and a tenfold enhanced bromide concentration. Based on the data in Fig. 1 and the surface coverage estimated from the XPS data, a constant ozonide complex coverage on the surface of $10^{12}$ complexes per cm² is assumed, so that the surface reactivity is the same for both scenarios. The size dependence of the bulk reactivity comes from the fact that the oxidation rate scales with the volume of the particle for smaller particles, whereas the uptake coefficient is the oxidation rate normalized to the surface area (Supplementary Note 1, Eq. 4a). Therefore, for environmental conditions, the surface reactivity may largely dominate over the bulk reactivity by two orders of magnitude, especially for brine

pockets of smaller dimensions or aerosol particles. Since the global distribution of BrO responds notably to the oxidation of bromide by ozone[5], and since our results here provide a significantly larger contribution by the surface reaction than the previously recommended parameterization[24], we expect notable changes to the relative importance of this reaction among the multiphase halogen chemical cycling reactions.

In summary, by combining kinetic studies with theoretical calculations and spectroscopy, we have demonstrated that the reactivity at the interface between bromide aqueous solutions and ozone from the gas phase is higher than in the bulk. While the kinetic experiments indirectly reconfirm and refine the picture of a precursor mediated process at the surface of aqueous solutions, the XPS results provide a clear spectroscopic evidence of this intermediate, the [Br•OOO⁻] pre-complex. Furthermore, a rough estimate of the surface coverage from the photoemission spectra of the Br 3$d$ at increasing information depths shows that the new species has a preference for the interface, in good agreement with first-principle MD simulations. This multi-method approach investigates an elusive reaction intermediate that has been predicted from theory and interpretation of kinetic data but never directly observed in experiments, to the best of our knowledge. In addition, this work shows its preference for the liquid–gas interface, and sheds light on mechanistic and structural aspects of the reaction. The results provide evidence for a stronger contribution of the surface oxidation of bromide than previously thought, which will require re-assessment of the impacts on the global ozone budget and mercury deposition[5]. In turn, the formation of ozonides on surfaces may be a widespread phenomenon and a key step of important oxidation processes relevant not only for atmospheric chemistry but also for the effects of atmospheric particles on human health[23, 25, 49].

## Methods

**Kinetic experiments**. Kinetic experiments were conducted in a flow reactor setup previously described by Lee et al.[21]. Briefly, the setup comprises of a temperature regulated Teflon trough (surface area = 102 cm²) on which 10 ml–45 ml of the reactive solution (NaBr (Sigma Aldrich) in deionized water) is loaded uniformly. Ozone is generated by UV light at different intensities from a mixture of 400 ml min⁻¹ O₂ and 600 ml min⁻¹ N₂. This gas flow is cooled and humidified to the water vapour pressure in the trough at the set temperature before delivery to the trough. Gas flow is alternated between a bypass to measure the maximum (initial) O₃ concentration and the trough to measure the O₃ left after reactive uptake by the solution. O₃ concentration was measured using a commercial ozone monitor (Teledyne API model 400).

**Theory**. Geometric optimization of the reaction intermediates were performed at MP2/6-311++g(df,p) level[50] while, to improve the energetic, single point energy calculations were employed at CCSD(T)/6-311++g(df,p) level[51] on the top of the optimized structures obtained with MP2. Electronic structure calculations were performed using Gaussian09[52].

First-Principle MD simulations were done as implemented in the CP2K code to study the stability and dynamics of the ¹[Br•OOO⁻] on the surface of liquid water[53]. The optimized geometry for ¹[Br•OOO⁻] obtained at MP2 level was placed on the top of an equilibrated water slab of 216 water molecules. The simulation was done under NVT conditions, at 300 K, using BLYP[54, 55] and Grimme dispersion correction[56], Goedecker-Teter-Hutter pseudopotentials[57] and DZVP basis set was employed in combination with plane wave representation for the valence electrons.

CEBE were calculated on top of MP2/6-311++g(df,p) geometries using GAMESS suite of codes[58]. For the CEBE in 5 water molecules cluster (Table 1), configurations were extracted from first-principle MD simulations. The core-hole state geometries were assumed to be identical to the corresponding ground states. The effect of relaxation was described by the ΔMP2 approach[59, 60] with the electron correlation described at MP2/aug-cc-pVTZ level of theory. The CEBE was defined as the difference between the ground state energy and the cation core-hole state formed by ejection of a 3$d$ electron from the Br⁻. Freezing the molecular orbitals during the SCF procedure prevented the collapse of the core-hole state to the more stable energetic state.

**Liquid jet XPS**. In-situ XPS was acquired at the near-ambient pressure photoemission endstation (NAPP), equipped with the liquid microjet setup.

Measurements were performed at the Surfaces/Interfaces: Microscopy (SIM) beamline of the Swiss Light Source (SLS) at the Paul Scherrer Institute (PSI). The electron analyzer uses a three-stage differentially pumped electrostatic lens system and a hemispherical analyzer to collect photoelectrons from samples in chamber pressures up to the mbar range[61]. For the present experiments, a quartz nozzle, forming a liquid microjet with a diameter of 24 μm, was used to deliver a 0.125 M aqueous solution of Br⁻ into the chamber at a flow rate of 0.35 ml min⁻¹. The liquid was cooled to 277 K in a pre-cooling coil located immediately before entry into the experimental chamber. Based on these parameters, and considering the working distance of the quartz nozzle with respect to the detection point, a 100 μs time can be estimated between the injection of the liquid and the detection point. Further technical details about the procedures adopted during the experiments can be found in the Supplementary Note 4.

During the experiment we made use of linearly polarized light at 0°, and set the photon energy for the detection of Br $3d$ to 350, 450 and 650 eV, resulting in photoelectron kinetic energies of 276, 376 and 576 eV, respectively. The binding energy scale of the spectra was calibrated using the O $1s$ gas phase peak of water (539.9 eV) as a reference[32]. The streaming potential of a 0.125 mol l⁻¹ bromide aqueous solution should be between 0.1 and 0.2 eV[62]. The binding energy of the O $1s$ peak corresponding to the condensed phase is in good agreement with the values reported in literature[32, 63] and did not change during the whole measurements. Other technical details about the experimental setup and the processing of the photoemission spectra can be found in the Supplementary Note 4.

**Data availability**. The data that support the findings of this study are available from the corresponding authors upon reasonable request.

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

## Acknowledgements

For computer time, this research used the resources of the Supercomputing Laboratory at King Abdullah University of Science and Technology (KAUST) in Thuwal, Saudi Arabia. The engineering work and technical support by Mario Birrer is greatly acknowledged. J.E. acknowledges the Swiss National Science Foundation (SNF), grant 155999. M.A., F.O. and S.C. acknowledge funding by the Swiss National Science Foundation (grants no 149492 and 169176). We thank Michel Rossi for providing the ozone generator.

## Author contributions

L.A., P.B.S., I.G. and M.A. conceived the project. L.A. coordinated and participated to synchrotron beamtimes, processed the XPS data, and drafted the manuscript. J.E. and A.G. performed the kinetic experiments, processed the data and drafted the section about kinetic measurements. F.O. contributed to the design of the liquid-jet experiment, participated to the synchrotron beamtimes, and contributed to the analysis of the XPS data. S.C., M.T.L., P.C. A., and A.K. contributed to the synchrotron beamtimes. T.B.R. helped in the processing of the kinetic measurements output. M.A. helped to process both the XPS and kinetic measurements data, also drafting the manuscript. M.V. employed the geometrical optimization and MP2 and CCSD levels. M.V. and I.G. computed the core electron binding calculations at MP2 level. M.A.C. and I.G. performed the first-principle MD, analyzing the trajectory results. I.G. and J.S.F. analyzed and drafted the MD and electronic structure calculations results. All the authors contributed in the discussion during the preparation of the final manuscript.

## Additional information

**Competing interests:** The authors declare no competing financial interests.

