## [Peer Review File · Nature Communications]

Reviewers' comments:

Reviewer #1 (Remarks to the Author):

The manuscript by Artiglia et al. presents an in-depth study of the generation of gaseous halide components from the aqueous phase, here with the focus on bromide oxidation at the air-liquid interface. Chlorine and bromine catalytic reactions are key processes in the atmospheric ozone depletion. This work aims to elucidate the formation and kinetics of the proposed $[\text{Br}\bullet\text{OOO-}]$ precursor in the reaction to gaseous hypobromous acid. The authors strive to illuminate the possible formation of this intermediate from multiple viewpoints by employing kinetics and photoelectron spectroscopic experiments as well as theoretical studies on the relevant electronic structure of the precursor complex and its surface propensity via dynamics simulations. The kinetic experiments and the theoretical parts are carried out and discussed in great detail. The XPS experimental part needs some clarifications, and I would be wary to put too much weight on the results. Overall, I recommend the publication of the article after some necessary modifications and additions. I will give comments and suggestions below.

1) Comment on ozone uptake and kinetics:

- The kinetics experiment in the flow reactor was discussed and analyzed in great detail, but I missed an appropriate discussion of the conditions and possible differences to the liquid jet experiment. It was only hinted that the timescales are large enough to establish equilibrium with the sentence "The continuously renewed surface of the flowing liquid jet avoids radiation damage effects but still allows probing a surface that is locally in equilibrium with the first few tens of nanometer of bulk aqueous phase" (page 5). I wonder if there are any effects of large ozone concentration gradients (before injection \rightarrow gas interaction nozzle \rightarrow analyzing chamber). Also, even at NAP pressures of 0.25 mbar the liquid jet is rapidly evaporating and cooling down, which may affect the surface abundance and stability of the precursor.

2) Comment on the electronic structure calculations:

- The inclusion of the energetic calculations to show the stabilizing effect of water is a nice addition. However, I think the section discussing these results (section 2 in the main text) does not add much value in its current form. This part should be discussed more in-depth and the connection to the experiments should be highlighted. For example, how realistic is the calculation with 1-to-4 water clusters without accounting for the liquid environment (polarization effect), even if this is meant just for partly solvated complexes near the surface? What is expected when a full solvation shell is formed? Is further stabilization or settlement of the current values expected? Are there any comparable theoretical or experimental studies for the water cluster calculations to refer to? It is difficult to judge for the reader what to take away from this section including Figure 2 besides qualitative statements.

3) Comments on XPS experimental results:

- The procedure how the data is handled and how the authors draw their conclusions is lacking or at least not well explained. This starts with the analysis of the Br core peaks. While there is a visible change in the ozone dosed spectrum, it is not clear how the extra peak doublet at apparently 0.7 eV shifted energy was assigned. What were the fit conditions and constraints? Are there possible alternate explanations?

- The assignment of the new feature as the proposed $[\text{Br}\bullet\text{OOO-}]$, which is done with measurements of reference solutions and core-level binding-energies (called CEBE in the manuscript) from MP2 calculations, is too readily made. It is written on page 8 "In agreement with the experimental results, the CEBE of the $[\text{Br}\bullet\text{OOO-}]$ complex is closer to that of the bromide than the CEBE of the hypobromite." But when looking at the table provided on page 9, I find the exact opposite, namely that $[\text{Br}\bullet\text{OOO-}]$ should be rather close to hypobromite. How is this discrepancy explained? Are the used model and made assumptions for the calculation appropriate? As I see it, without initial knowledge that there can only exist $[\text{Br}\bullet\text{OOO-}]$ under these experimental conditions, it will be very difficult to unequivocally identify the species just from the XPS spectra. In any case, the discussion should be more critical to rule out other possibilities.

- I hope the authors are aware of the problem streaming potentials caused by flowing liquid jets (e.g., Kurahashi et al., JCP 140, 174506, 2014). Such potentials may shift photoelectron energies

of gas (used as reference here) and liquid contributions depending on the salt content and flow conditions. This should be included in the discussion when comparing binding energies with theory.

- Unfortunately, the depth profiling results are only qualitative. I find it a missed opportunity to only measure the ratios at three energies in a rather narrow range (in the chosen energy range of 276 eV to 575 eV the mean escape depth may only change from 15 Å to about 25 Å). More points at especially higher energies would have given a clearer picture. I assume that technical limitations are at play here. Together with above comment, I suggest to tone down this sentence a bit: "the analysis of the photoemission spectra of the Br 3d at increasing information depths shows that the new species has a strong preference for the interface" (page 13).
- While I appreciate the effort to quantify the escape depth and the surface coverage of [Br•OOO-] from the available depth profiling data (part 3.3 in the SI), I would be cautious to assign too much meaning to the obtained values. There is too much uncertainty involved considering the fact that the results is mostly determined by only one data point (at high MED) and the overall rather bad fit of the model to the data.

Reviewer #2 (Remarks to the Author):

This paper represents outstanding science on a difficult problem, the identification of surface intermediates on aqueous solutions. The combination of sophisticated jet XPS experiments and theory is very convincing regarding identification of BrOOO at the surface, and its enhanced reactivity in that region. As they point out, BrOOO has been proposed to be formed in the bulk but to this reviewer's knowledge, this is the first identification at the surface.

They point out at the beginning of the Introduction and Discussion that such chemistry is important in the atmosphere, but also that it is of interest in other fields. The latter seems a little nebulous so if they could give some specific examples, it might strengthen the importance of this work in terms of the Nature Communications audience. It might also help if they could also comment on whether this is likely to change the results of atmospheric models by any significant amount.

Minor points:

1. Page 5, 2nd sentence from top starting with "More recent evidence from liquid jet XPS data..." but ref 27 cited seems to be a theory paper, not experimental (although it is interpreting the XPS data).
2. Figure 1b shows predictions for mixtures of chloride and bromide, and is discussed later in the paper. It would seem more logical to show the results for the bromide solution alone to compare to Fig 1a, maybe put Fig 1b in its own figure (or in SI)?
3. Bottom of page 6 talks about spin crossing to the triplet PE surface. It might help the reader to show this on Fig 2 somehow?
4. Figure 3 caption: Should part c say "deconvolution of the raw spectra of Fig 3b" (not Fig 3a)?
5. Page 11, top paragraph: The estimate of $2E_{12}$ complexes cm^{-2} presumably depends on the O₃ concentration? If so, what O₃ concentration does this estimate apply to?

Dear editor

We would like to thank the two reviewers for their constructive comments on our manuscript. Point by point responses to each comment as well as the actions taken in the manuscript are provided below in italics.

Reviewer #1 (Remarks to the Author):

The manuscript by Artiglia et al. presents an in-depth study of the generation of gaseous halide components from the aqueous phase, here with the focus on bromide oxidation at the air-liquid interface. Chlorine and bromine catalytic reactions are key processes in the atmospheric ozone depletion. This work aims to elucidate the formation and kinetics of the proposed [Br•OOO-] precursor in the reaction to gaseous hypobromous acid. The authors strive to illuminate the possible formation of this intermediate from multiple viewpoints by employing kinetics and photoelectron spectroscopic experiments as well as theoretical studies on the relevant electronic structure of the precursor complex and its surface propensity via dynamics simulations. The kinetic experiments and the theoretical parts are carried out and discussed in great detail. The XPS experimental part needs some clarifications, and I would be wary to put too much weight on the results. Overall, I recommend the publication of the article after some necessary modifications and additions. I will give comments and suggestions below.

Reply:

We would like to thank the reviewer for praising our work and recommending publication of the article. Moreover, we are grateful to the thoughtful evaluation and constructive comments, which helped us to address the requested modifications and additions. Before answering to the specific comments, we wish to point out that while the XPS results do not provide quantitative information with respect to absolute concentrations at the precision level possible in the kinetics experiments, they provide a powerful and chemically selective verification of the mechanism in line with the theoretical calculations and in agreement with the mechanistic interpretation of the kinetic data. Reaction intermediates involved in the heterogeneous kinetics atmospheric chemical systems have never been caught in the act as in this study. Detailed point by point responses and the actions taken on the manuscript are described below.

1) Comment on ozone uptake and kinetics:

- The kinetics experiment in the flow reactor was discussed and analyzed in great detail, but I missed an appropriate discussion of the conditions and possible differences to the liquid jet experiment. It was only hinted that the timescales are large enough to establish equilibrium with the sentence “The continuously renewed surface of the flowing liquid jet avoids radiation damage effects but still allows probing a surface that is locally in equilibrium with the first few tens of nanometer of bulk aqueous phase” (page 5). I wonder if there are any effects of large ozone concentration gradients (before injection -> gas interaction nozzle -> analyzing chamber). Also, even at NAP pressures of 0.25 mbar the liquid jet is rapidly evaporating and cooling down, which may affect the surface abundance and stability of the precursor.

Reply:

According to this comment, we analyzed in depth the experimental conditions adopted during the liquid jet experiment. Our reply is reported below, divided into sections describing different but connected issues:

i) surface – bulk liquid equilibration: the time scale for diffusion is such that if a chemical or physical process would deplete a major fraction of a main surface component at once, diffusion from the bulk would re-establish equilibrium within a few microseconds if this component is available at 0.1 M in the bulk.

Actions taken: an estimate of the time required for surface-bulk equilibration is already provided in the first paragraph of the discussion section on page 12-13. We now additionally provide the calculation based on Lee et al. (2015) and Winter et al. (2006) in more detail in the SI. The following text was added on page 16-17 of the SI:

“The evolution of the surface concentration of an initially free surface by diffusion from the bulk with time is given by equation 12:

$$c_s = 2 \left(\frac{D}{\pi} \right)^{1/2} ct^{1/2}$$

where c is the bulk concentration of the bromide solution ($7.53 \cdot 10^{19}$ molecules/cm³), c_s is the surface concentration of [Br•OOO•] (estimated as $2 \cdot 10^{12}$ molecules/cm²), and D is the diffusion coefficient ($2.41 \cdot 10^{-10}$ cm²/s), evaluated through the Stokes-Einstein equation. The time needed to re-establish the equilibrium concentration is approx. 2 μs, much lower than the time between the injection of the liquid filament and the detection point (100 μs).”

ii) gas – liquid equilibration: as in (i), the diffusion in the liquid is fast enough that gas –liquid equilibrium over the probe depth of the XPS experiments (a few nm) is established within 1 μs. Because the solubility of O₃ is low, the net flux into the liquid is low, and no gradients will limit this flux in the gas phase. Since the residence time of the liquid filament within the dosing system is approx. 100 μs and between the dosing system and the point at which it hits the X-ray is 100 μs, the gas – surface and gas – liquid equilibria follow the changing pressure in the gas phase. Unfortunately the pressure inside the dosing system cannot be measured. We plan to run calculations to evaluate it and its role on the stability of the liquid filament.

A preliminary experiment carried out without the gas dosing system showed a similar evolution of the Br 3d spectrum while dosing ozone (0.1-0.2 mbar background pressure). However, the effect was less pronounced than after employing the gas dosing system, indicating that loss of O₃ to the chamber walls has a significant effect on the O₃ exposure on the liquid.

Actions taken: we added a paragraph discussing the gas-liquid equilibrium on page 16 of the SI. The text is the following:

“The diffusion of the gas into the liquid is fast enough that gas –liquid equilibrium over the probe depth of the XPS experiments is established within 1 μ s. Because the solubility of ozone is low, the net flux into the liquid is low, and no gradients will limit this flux in the gas phase. The residence time of the liquid filament within the dosing system is approx. 100 μ s and between the dosing system and the point at which it hits the X-ray is 100 μ s. Therefore, the gas – surface and gas – liquid equilibria follow the changing pressure in the gas phase. A preliminary experiment carried out without the gas dosing system showed a similar evolution of the Br 3d spectrum while dosing ozone (0.1-0.2 mbar background pressure). However, the effect was less pronounced than after employing the gas dosing system, indicating that loss of ozone to the chamber walls has a significant effect on the gas exposure on the liquid.”

iii) cooling of the jet: as explained in the text (Methods), the solution was thermalized at 277 K before being injected into the experimental chamber by means of the nozzle. Due to the near ambient pressure conditions (the pressure during the experiment was set to 0.25 mbar), the liquid filament undergoes evaporation and, consequently, cools down. It is difficult to quantify the entity of such a temperature change. Thanks to the design of liquid microjets, by which the liquid filament is injected at a high speed and has laminar flow in the probed region (the liquid filament is hit by the photon beam 100 μ s after its injection in NAP), relevant changes in the physical properties of the solution can be ruled out. Although the temperature of the solution may decrease by a few K, we do not expect an appreciable change of the surface abundance of the ozonide, because the reaction has a weak dependence on the temperature.

iv) comparison to flow reactor experiments: these experiments were performed at 274 K to roughly match with the temperature in the liquid jet XPS experiments. In turn, the kinetic experiments operate at much longer time scales, i.e., seconds. However, the kinetic model that explains the kinetic results is based on a classic Langmuir-Hinshelwood approach, in which the adsorbed precursor, the ozonide in this case, limits the reaction under conditions of low O₃ concentration. Supported by the theoretical calculations, the formation of this intermediate is very fast, so that we see its steady state coverage after the μ s exposure time scales of the liquid jet experiment, while this same coverage drives the steady state kinetics observed in the flow reactor.

2) Comment on the electronic structure calculations:

- The inclusion of the energetic calculations to show the stabilizing effect of water is a nice addition. However, I think the section discussing these results (section 2 in the main text) does not add much value in its current form. This part should be discussed more in-depth and the connection to the experiments should be highlighted. For example, how realistic is the calculation with 1-to-4 water clusters without accounting for the liquid environment (polarization effect), even if this is meant just for partly solvated complexes near the surface? What is expected when a full solvation shell is formed? Is further stabilization or settlement of the current values expected? Are there any comparable theoretical or experimental studies for the water cluster calculations to refer to? It is difficult to judge for the reader what to take away from this section including Figure 2 besides qualitative statements.

Reply: *We thank the reviewer for his comment and we agree that more discussion in section 2 of the main text could be helpful. Modeling the physicochemical behavior of surfactants at the liquid water interface is computationally challenging, especially if expensive levels of theory are needed, such as*

MP2, CCSD, CCSD(T) (used in this work). Continuum model could be used to account approximately the full solvation shell, assuming a homogenous environment (e.g., bulk water). However, at the best of our knowledge, there are no continuum models that can consider the inhomogeneity of the interfacial environment.

For this reason, a water cluster is a practical way to describe the solvation environment of the reaction complexes at interface. If the number of water molecules is sufficiently large, the cluster provides a reasonable approximation of the interfacial environment. Even if there is not a general rule to evaluate the required minimum number of water molecules, combination of first principle MD, mixed QM-MM methods and electronic structure calculations have shown that 3-4 molecules are sufficient to give the critical behavior at the interface for several reaction systems. [Gerber et al., *Acc. Chem. Res.* 2015, DOI: 10.1021/ar500431g; Inaba, *J. Phys. Chem. A* 2014, DOI: 10.1021/jp5021406]. This is also our case. In Figure S5 (see the SI) we included the energy difference between $^1\text{O}_3 + ^1[\text{Br}^- \bullet \text{xH}_2\text{O}]$ and $^1[\text{Br}^- \bullet \text{OOO} \bullet \text{nH}_2\text{O}]$ with $n=1,2,3,4$, which were not present before. The energy differences are: -23.6 kJ/mol, -6.1 kJ/mol, -10.5 kJ/mol, -5.7 kJ/mol for $n=1,2,3,4$, water, respectively. This energy difference can be considered converged within the chemical accuracy of the method (< 1 kcal/mol, Ramabhadran & Raghavachari, *JCTC*, 2013). Concerning other comparable theoretical or experimental studies, there are several studies on ozone-water clusters (e.g., J. M. Anglada et al., *JPC-A*, 2013 and references herein) but, at the best of our knowledge, this is the first study on the bromide-ozone reaction in water cluster.

Actions taken: Figure 4a shows now results from a longer DFT-MD trajectory (now 8.5 ps), to prove the stability of the pre-complex at the interface. Interestingly, we see that after ca 6 ps there is a reorientation of O_3 in the pre-complex. This is also quite solid proof its interfacial stability, otherwise one could expect O_3 to leave after reorientation if that was energetically feasible.

The main text was modified as follows:

On page 7, Section 2 (first paragraph)

“Previous electronic structure calculations for the oxidation of bromide by ozone in the gas phase showed that the reaction is activated by the formation of a stable $^1[\text{Br} \bullet \text{OOO}]$ pre-complex likely followed by a spin crossing to the triplet potential energy surface due to the different spin state of the products⁹. Herein, additional electronic structure calculations were performed to address the influence of water on the stability of the $^1[\text{Br} \bullet \text{OOO}]$ pre-complex. Water can profoundly affect the reaction rates and the nature of different atmospheric (and non-) reactions by coordinating (hydrogen bonds) to the reagents, products and the different reaction pre-complexes (new reference 32). Reaction profiles in solvated environment are often remarkably different from the ones obtained in the gas phase (new references 33, 34). Water cluster has been successfully used to model reactions in aqueous solutions, the surface of water, ice and aerosols (new references 33, 34, 35, 36, 37, 38, 39). If a sufficient number of water molecules is included in the electronic structure calculations, water cluster has been proven to describe reasonably the solvation environment of different reaction systems (new references 40, 41, 42). In our case, Figure S5 in the supplemental material shows how the energy difference between reaction complexes converge within the chemical accuracy of the method⁴³ for cluster of four water molecules. The water cluster approach has the

advantage of keeping the system size small, allowing the use of high-level theory electronic structure calculations. Figure 2 shows the reaction profile with 4 water molecules, using the singlet ground state of the reactants as an initial and reference level for the energetics. Comparing this reaction profile with previous gas phase results, we conclude that water further stabilizes the pre-complex with respect to the reactants and the other reaction complexes.”

On page 10, paragraph 4:

“First principle MD simulations, which are a particularly suitable tool to study the dynamics and stability of non-standard compounds, were used to address the bulk vs. surface propensity of the different reaction intermediates³⁵. Figure 4a shows a snapshot (corresponding to 8.5 ps) of the MD trajectory of the pre-complex on the surface of a water slab at 300 K. The inset shows the distance between bromine and each of the oxygen atoms of $^1[\text{Br}\bullet\text{OOO}^-]$ along the MD trajectory. This distance fluctuates around the average value of 2.7 Å, which is consistent with that obtained for the optimized geometries by electronic structure calculations (see ref. 9 and SI, section 2.1). This further supports the scenario of a pre-complex stabilized on the surface of liquid water at finite temperature. Moreover, Figure 4b shows the density profile of the Br and OOO groups in the $^1[\text{Br}\bullet\text{OOO}^-]$ intermediate position along the coordinate perpendicular to the water interface, confirming that $^1[\text{Br}\bullet\text{OOO}^-]$ remains at the interface during the whole trajectory, with the Br group close to the OOO group.

3) Comments on XPS experimental results:

- The procedure how the data is handled and how the authors draw their conclusions is lacking or at least not well explained. This starts with the analysis of the Br core peaks. While there is a visible change in the ozone dosed spectrum, it is not clear how the extra peak doublet at apparently 0.7 eV shifted energy was assigned. What were the fit conditions and constraints? Are there possible alternate explanations?

Reply: We thank the reviewer for this comment. In good agreement with previous results published in *J. Phys. Chem. A* 2015, 119, 4600, the photoemission spectra were fitted using Shirley background subtraction and Gaussian line shapes. For Br 3d, the spin-orbit split was fixed at 1.03 eV. The Full width at half maximum (FWHM) of the peaks was constrained for the whole set of processed data (1.05 eV). A single doublet allowed getting a good deconvolution of the Br 3d signals acquired without dosing and while dosing oxygen. In the presence of ozone a second doublet had to be added to reach the same level of correlation. The second doublet is made of Gaussian peaks having a spin-orbit split fixed at 1.09 eV and FWHM of 1.10 eV. The best correlation was obtained applying a chemical shift of +0.7 eV to the binding energy of the second doublet. To the best of our knowledge, other explanations for the observed modifications of the Br 3d after introducing ozone in the gas mixture cannot be found. The experimental conditions were not changed for the whole duration of the experiment. The liquid flow and the pressure were constant, and ozone was introduced by switching on a generator in line with the oxygen delivery gas line. The Swiss light source synchrotron operates in top-up mode, thus the photon flux did not change during measurements. It is well known from the literature that x-ray beams can induce the radiolysis of water, leading to the formation of hydroxyl radicals (*Journal of Synchrotron Radiation* 2012, 19, 875 and references therein). Thanks to

the high speed of the injected liquid filament, the liquid microjet technique limits the beam damage, and the concentration of radicals does not reach $1 \cdot 10^{-6}$ mol/L under the experimental conditions adopted in this study.

Actions taken: *the fit conditions and constraints have been specified in section 3.1 of the supplementary information. The following paragraphs were added to the SI (section 3.1):*

“The Swiss light source synchrotron operates in top-up mode, thus the photon flux did not change during measurements. It is well known from the literature that x-rays can induce the radiolysis of water, leading to the formation of highly reactive hydroxyl radicals (reference 34 and references therein), which could affect the measurements. Thanks to the high speed of the injected liquid filament, the liquid microjet technique limits the beam damage, and the concentration of radicals does not reach $1 \cdot 10^{-6}$ mol/L under the experimental conditions adopted in this study.”

“The photoemission spectra were fitted using Shirley background subtraction and Gaussian line shapes⁴. For Br 3d, the spin-orbit split was fixed at 1.03 eV. The Full width at half maximum (FWHM) of the peaks was constrained for the whole set of processed data (1.05 eV). A single doublet allowed getting a good deconvolution of the Br 3d signals acquired without dosing and while dosing oxygen. In the presence of ozone, a second doublet had to be added to reach the same correlation. The second doublet is made of Gaussian peaks having a spin-orbit split fixed at 1.09 eV and FWHM of 1.10 eV. The best correlation was obtained applying a chemical shift of +0.7 eV to the binding energy of the second doublet.”

- The assignment of the new feature as the proposed $[\text{Br}\bullet\text{OOO}^-]$, which is done with measurements of reference solutions and core-level binding-energies (called CEBE in the manuscript) from MP2 calculations, is too readily made. It is written on page 8 “In agreement with the experimental results, the CEBE of the $[\text{Br}\bullet\text{OOO}^-]$ complex is closer to that of the bromide than the CEBE of the hypobromite.” But when looking at the table provided on page 9, I find the exact opposite, namely that $[\text{Br}\bullet\text{OOO}^-]$ should be rather close to hypobromite. How is this discrepancy explained? Are the used model and made assumptions for the calculation appropriate? As I see it, without initial knowledge that there can only exist $[\text{Br}\bullet\text{OOO}^-]$ under these experimental conditions, it will be very difficult to unequivocally identify the species just from the XPS spectra. In any case, the discussion should be more critical to rule out other possibilities.

Reply: *We agree with the reviewer that the calculated CEBE of the $[\text{Br}\bullet\text{OOO}^-]$ is closer to the one of hypobromite than to the one of bromide. Nevertheless, when water molecules are added to the cluster, the energy difference between the CEBE of $[\text{Br}\bullet\text{OOO}^-]$ and hypobromite increases (1.0 and 0.86 eV with 3 and 5 water molecules, respectively). At the same time, the energy difference between the $[\text{Br}\bullet\text{OOO}^-]$ and the bromide decreases (from 2.78 to 1.81 eV). This suggests that the solvation of the species has a fundamental role. The theoretical evaluation of the CEBE also suggests that the sequence of the CEBE is $\text{CEBE}(\text{Br}^-) < \text{CEBE}([\text{Br}\bullet\text{OOO}^-]) < \text{CEBE}(\text{BrO}^-)$, in agreement with the experimental results.*

It is important to highlight that the CEBE are calculated for the species solvated in small water clusters (up to 5 water molecules). As mentioned above, water clusters is a reasonable approximation for interfacial environment but may not reproduce well chemical species that are fully solvated in the bulk. Our results strongly support the presence of $[\text{Br}\bullet\text{OOO}^-]$ at the interface but we do not know yet

the exact solvation environment for hypobromite (bulk vs. surface). For this reason, the calculated CEBE for BrO^- should be taken with some caution.

Actions taken: we modified the text on page 8 as follows:

“To identify the new spectral feature, we acquired the Br 3d spectra of two reference aqueous solutions of possible oxidation products, i.e. 0.08 M hypobromite and 0.125 M bromate (Figure 3d). As expected, the higher the oxidation state of bromine, the larger the positive shift of the binding energy. A chemical shift of +2.1 eV is observed for hypobromite, and of +7.0 eV for bromate. None of them corresponds to that of the new doublet. It is well known that x-rays can induce the radiolysis of water³⁵, leading to the production of reactive hydroxyl radicals that may react with bromide ions. All the Br 3d spectra were recorded under the same experimental conditions (excitation energy, photon flux), and the high speed of the liquid filament ensures that the concentration of photo-generated hydroxyl radicals remains below $1.0 \cdot 10^{-6}$ mol/L. Therefore beam damage can be excluded. In parallel to the XPS data, we calculated the core electron binding energy (CEBE) at MP2/aug-cc-pvtz theory level of both the structure obtained at MP2/6-311++(df,p) geometric optimization and from first-principle MD (Table 1). It is important to highlight that the CEBE are calculated for the species solvated in small water clusters. This reproduces quite well an interfacial environment, but may not reproduce well species that are fully solvated in the bulk. As compared to the CEBE of gas phase species, when water is added the Δ between the bromide and the $[\text{Br}\bullet\text{OOO}^-]$ decreases, whereas that between the $[\text{Br}\bullet\text{OOO}^-]$ and the hypobromite increases. This suggests that the solvation sphere has a fundamental role. At the same time, theoretical calculations reproduce the same sequence of CEBE showed by XPS, i.e. $\text{CEBE}(\text{Br}^-) < \text{CEBE}([\text{Br}\bullet\text{OOO}^-]) < \text{CEBE}(\text{BrO}^-)$. In summary, the combination of in-situ XPS and theoretical calculations provides strong indications for the formation of an ozonide complex.”

- I hope the authors are aware of the problem streaming potentials caused by flowing liquid jets (e.g., Kurahashi et al., JCP 140, 174506, 2014). Such potentials may shift photoelectron energies of gas (used as reference here) and liquid contributions depending on the salt content and flow conditions. This should be included in the discussion when comparing binding energies with theory.

Reply: we thank the reviewer for this comment. We choose the bromide concentration (0.125 mol/L) to get enough intensity of the photoemission signal while keeping the streaming potential reasonably low. In their paper, Kurahashi et al. measured the largest streaming potentials for NaBr solutions in the 5-10 mM concentration range. At a Br^- concentration of 0.125 mol/L, the streaming potential should be between 0.1 and 0.2 V (although the flow rate used by Kurahashi is 0.5 mL/min versus 0.35 used in our study). As specified on page 14 of the manuscript, we used the gas phase peak of water (539.9 eV) as a reference to evaluate the binding energy scale of the spectra. By using this reference, neither a shift of the peak of O 1s associated to the condensed phase nor a shift of the 3d peaks associated to the bromide was observed in the whole dataset. The binding energy of the O 1s peak of condensed water is in good agreement (± 0.1 eV) with previous literature records (B. Winter, et al. J. Chem. Phys. 2007, 126, 124504; B. Winter, M. Faubel Chem. Rev. 2006, 106, 1176). The binding energy scale was cross-checked using the $1b_1$ peak of the valence band, whose binding energy was calibrated in agreement with the value provided by Kurahashi et al.

Actions taken: we modified the text on page 16 as follows:

“During the experiment we made use of linearly polarized light at 0°, and set the photon energy for the detection of Br 3d to 350, 450 and 650 eV, resulting in photoelectron kinetic energies of 276, 376 and 576 eV, respectively. The binding energy scale of the spectra was calibrated using the O 1s gas phase peak of water (539.9 eV) as a reference²⁸. The streaming potential of a 0.125 mol/L bromide aqueous solution should be between 0.1 and 0.2 eV⁵⁰. The binding energy of the O 1s peak corresponding to the condensed phase is in good agreement with the values reported in literature^{28, 51} and did not change during the whole measurements. Other technical details about the experimental setup and the processing of the photoemission spectra can be found in the SI.”

- Unfortunately, the depth profiling results are only qualitative. I find it a missed opportunity to only measure the ratios at three energies in a rather narrow range (in the chosen energy range of 276 eV to 575 eV the mean escape depth may only change from 15 Å to about 25 Å). More points at especially higher energies would have given a clearer picture. I assume that technical limitations are at play here. Together with above comment, I suggest to tone down this sentence a bit: “the analysis of the photoemission spectra of the Br 3d at increasing information depths shows that the new species has a strong preference for the interface” (page 13).

Reply: *We share the opinion of the reviewer about our attempt to quantify the amount of precursor at the surface. Quantification of the complex from the Br 3d spectra is relying on high signal to noise ratio, which in turn is a function of the photon flux, the analyzer pass energy, the number of sweeps acquired over a period of stable liquid jet operation, all within limited amounts of beamtime at the synchrotron. Overall efficiency in this regard was highest for the 250-550 eV photoelectron kinetic energy range that promises highest sensitivity for surface to bulk contrast and allows making use of the maximum of the flux curve of the corresponding beamline and keeping the pass energy of the spectrometer (50 eV) constant.*

Actions taken: *we modified the text on page 14 to better express the uncertainty:*

“Furthermore, a rough estimate of the surface coverage from the photoemission spectra of the Br 3d at increasing information depths shows that the new species has a preference for the interface, in good agreement with first-principle MD simulations.”

- While I appreciate the effort to quantify the escape depth and the surface coverage of [Br•OOO-] from the available depth profiling data (part 3.3 in the SI), I would be cautious to assign too much meaning to the obtained values. There is too much uncertainty involved considering the fact that the results is mostly determined by only one data point (at high MED) and the overall rather bad fit of the model to the data.

Reply: *we agree with the reviewer. More experimental points acquired at higher photoelectron kinetic energy would have made the depth profile more reasonable. Our aim was only to provide the reader with a rough estimate of the surface coverage of [Br•OOO-] and to have a proof of its preference for the interface.*

Actions taken: In agreement with the previous comment, we toned down the sentence commenting the result of our depth profile (page 14 of the manuscript). We also modified the first sentence of section 3.3 (SI) as follows:

“Supplementary Figure 7 shows a rough estimate of the surface coverage of [Br•OOO] complex based on the XPS data acquired at increasing photoelectron kinetic energy (276, 376, and 576 eV).”

Reviewer #2 (Remarks to the Author):

This paper represents outstanding science on a difficult problem, the identification of surface intermediates on aqueous solutions. The combination of sophisticated jet XPS experiments and theory is very convincing regarding identification of BrOOO at the surface, and its enhanced reactivity in that region. As they point out, BrOOO has been proposed to be formed in the bulk but to this reviewer's knowledge, this is the first identification at the surface.

They point out at the beginning of the Introduction and Discussion that such chemistry is important in the atmosphere, but also that it is of interest in other fields. The latter seems a little nebulous so if they could give some specific examples, it might strengthen the importance of this work in terms of the Nature Communications audience. It might also help if they could also comment on whether this is likely to change the results of atmospheric models by any significant amount.

Reply:

We would like to thank this reviewer for sharing his excitement about our results saying that our work represents outstanding science on a difficult problem as well as for the constructive comments. We feel that the atmospheric chemistry focus of this work is of substantial interest to also a wider audience, since halogen chemistry is related to crucial aspects in atmospheric, climate, ecosystem and human health. While this is documented in the literature cited, we have slightly amended the text to reflect this better. In addition, we have also mentioned the fact that ozone treatment is a common technology in waste water clean-up. With respect to the atmospheric implications, we show Figure 1b, now moved to the end of the manuscript and converted into Figure 5, to express the relative importance of surface (as derived from this study) and bulk phase chemistry. We have emphasized more clearly in the revised version that this is a crucial aspect in the global atmosphere by better explaining the link to a recent model study on this subject (Schmidt et al. (2016) reference).

Actions taken: We modified the first and fourth paragraphs in the introduction as follows:

“In atmospheric chemistry, halogen atoms are important catalysts for ozone depletion both in the stratosphere and in the troposphere. In the stratosphere, halogen atoms are mainly formed from photolysis of anthropogenic halogenated hydrocarbons. In the troposphere, they result from photolysis of both organic and inorganic halogen compounds^{1, 2}. Due to the photochemical properties of these precursors, in the stratosphere, mainly chlorine and bromine are involved in ozone depletion, while in the troposphere, bromine, and iodine play more important roles³. Halogen atoms are not only relevant for the ozone budget, but are also oxidants themselves and are implicated in the deposition of mercury^{4,5} (added new reference 5). Halogen atoms are also intermediates in waste water treatment, where halogenated organic secondary products are of concern.”

“Oxidation of bromide by ozone had been studied since long due to its relevance in debromination of waste water^{8,9,10} (new reference 9 and 10). In the bulk aqueous phase and at neutral pH this reaction is slow¹¹. The addition of an acid promotes it by assisting the formation of hypobromite^{12, 13, 14}.”

- Figure 1b was removed from Figure 1 and converted into a new figure (Figure 5, page 14). Figure 5 is now discussed in the Discussion section (on page 13 and 14). The following parts were added:

“Under atmospheric conditions, the relative importance of bulk and surface reactivity needs a more careful analysis. In the environment, salt brine may occur in various forms and spanning a large size range in snow, on sea-ice, on frost flowers, in salt pans, or as sea spray aerosol particles. Figure 5 shows the bulk reactivity (in absence of surface reaction) and the surface reactivity according to the mechanism in this study, for a hypothetical spherical sea salt brine droplet or aerosol particle as a function of the diameter, for both a bromide concentration as in sea water and a tenfold enhanced bromide concentration. Based on the data in Figure 1 and the surface coverage estimated from the XPS data, a constant ozonide complex coverage on the surface of 1012 complexes/cm² is assumed, so that the surface reactivity is the same for both scenarios. The size dependence of the bulk reactivity comes from the fact that the oxidation rate scales with the volume of the particle for smaller particles, whereas the uptake coefficient is the oxidation rate normalized to the surface area (see SI section 1.2, equation 4a). Therefore, for environmental conditions, the surface reactivity may largely dominate over the bulk reactivity by two orders of magnitude, especially for brine pockets of smaller dimensions or aerosol particles. Since the global distribution of BrO responds notably to the oxidation of bromide by ozone 6, and since our new results provide a significantly larger contribution by the surface reaction than the previously recommended parameterization 24, we expect notable changes to the relative importance of this reaction among the multiphase halogen chemical cycling reactions.”

- The following sentence was added at the end of the manuscript:

“The results provide evidence for a stronger contribution of the surface oxidation of bromide than previously thought, which will require re-assessment of the impacts on the global ozone budget and mercury deposition⁶. In turn, the formation of ozonides on surfaces may be a widespread phenomenon and a key step of important oxidation processes relevant not only for atmospheric chemistry, but also for the effects of atmospheric particles on human health^{23, 25, 49}.”

Minor points:

1. Page 5, 2nd sentence from top starting with "More recent evidence from liquid jet XPS data..." but ref 27 cited seems to be a theory paper, not experimental (although it is interpreting the XPS data).

Actions taken: We thank the reviewer for this comment. We modified the sentence (page 5 of the manuscript) as follows:

“Recent MD simulations predicting the photoemission signal intensity by means of photoelectron scattering calculations, and liquid jet XPS data indicate a less pronounced surface enhancement,

which is in better agreement with the overall positive surface tension change (and thus negative surface excess) of bromide solutions^{27, 28}.”

2. Figure 1b shows predictions for mixtures of chloride and bromide, and is discussed later in the paper. It would seem more logical to show the results for the bromide solution alone to compare to Fig 1a, maybe put Fig 1b in its own figure (or in SI)?

Figure 1b has been moved to the end of the manuscript (and converted into Figure 5) to accompany the discussion section, where it is actually discussed, also in view of the general response regarding atmospheric implications above.

3. Bottom of page 6 talks about spin crossing to the triplet PE surface. It might help the reader to show this on Fig 2 somehow?

Reply: *Based on the energetics, there is a spin crossing between the reaction pre-complexes and the transition states, as shown by the crossing of the red and blue arrow in Figure 2. Nevertheless, the determination of the exact position of this spin-crossing (and of the relevant reaction coordinates) is very challenging and not within the scope of this paper, which is more focused on the stability of the reaction pre-complex.*

Actions taken: *We rephrased the caption of Figure 2 as follows:*

“Energetic profile for the bromide-ozone reaction with 4 water molecules along the singlet (blue) and triplet (red) surface obtained by CCSD(T)/6-311++G(df,p)//MP2/6-311++G(df,p) electronic structure calculations. All energies, which include the Zero Point Energy (ZPE) correction, are relative to the singlet reactants and are reported in kJ/mol. The spin crossing between the two potential energy surface is highlighted by the intersection of the red and blue arrows between the pre-reaction complexes and the transition states.”

4. Figure 3 caption: Should part c say "deconvolution of the raw spectra of Fig 3b" (not Fig 3a)?

Actions taken: *We modified the caption of Figure 3 according to the comment of the reviewer.*

5. Page 11, top paragraph: The estimate of $2E12$ complexes cm^{-2} presumably depends on the O_3 concentration? If so, what O_3 concentration does this estimate apply to?

Reply: *Based on the pressure measured in the experimental chamber and the conversion of oxygen to ozone (ca. 1.0%), and a Henry's law constant of ozone of around $10^{-2} \text{ M atm}^{-1}$ the concentration of ozone is approx. $2.5 \cdot 10^{-8} \text{ mol/L}$. As discussed later in the discussion section, the kinetic data indicate that the surface coverage of the precursor seems to be rather insensitive to the O_3 concentration likely due to saturation. However its value cannot be constrained from the kinetic data, as mentioned in the text. Therefore, no further discussion / speculation was added to the discussion.*

Actions taken: *we added the concentration of ozone to the sentence on page 11.*

REVIEWERS' COMMENTS:

Reviewer #1 (Remarks to the Author):

The authors have responded to the referee comments properly, and I now recommend publication of the manuscript. On page 3, second paragraph, hypobromous acid (HBr) should be (HOBr).

Reviewer #2 (Remarks to the Author):

The authors have answered my comments and suggestions satisfactorily and I recommend publication.

Dear editor

We would like to thank the two reviewers for their positive assessment of our revised version.

Reviewer #1 (Remarks to the Author):

The authors have responded to the referee comments properly, and I now recommend publication of the manuscript. On page 3, second paragraph, hypobromous acid (HBr) should be (HOBr).

Reply:

This typo has been corrected. At this point we have also made sure to be consistent about mentioning hypobromous acid and hypobromite, the conjugate base, as appropriate for the context. Since we have written the chemical mechanism formally with the acids protonated, the Introduction section now only mentions HOBr.

Reviewer #2 (Remarks to the Author):

The authors have answered my comments and suggestions satisfactorily and I recommend publication.

Reply:

We would like to thank this reviewer for his positive assessment of our revisions.

Actions taken: *No further specific actions taken.*